# Green Synthesis of Na abietate Obtained from the Salification of *Pinus elliottii* Resin with Promising Antimicrobial Action

**DOI:** 10.3390/antibiotics12030514

**Published:** 2023-03-04

**Authors:** Aline B. Schons, Patrícia Appelt, Jamille S. Correa, Mário A. A. Cunha, Mauricio G. Rodrigues, Fauze J. Anaissi

**Affiliations:** 1Department of Chemistry, Universidade Estadual do Centro-Oeste, Guarapuava 85040-167, Brazil; 2Department of Chemistry, Universidade Tecnológica Federal do Paraná, Pato Branco 85503-390, Brazil; 3Instituto Federal Catarinense, IFC, Campus Camboriú, Camboriú 88340-055, Brazil

**Keywords:** resin, abietic acid, DFT, antimicrobial, resistant bacteria, circular economy

## Abstract

The growing concern about the emergence of increasingly antibiotic-r4esistant bacteria imposes the need to search and develop drugs to combat these microorganisms. This, combined with the search for low-cost synthesis methods, was the motivation for the elaboration of this work. Abietic acid present in the resin of *Pinus elliotti var. elliotti* was used to generate a sodium salt by salification. The synthesis route was low-cost, consisting of only two reaction steps at mild temperatures without toxic organic solvents, and eco-friendly and easy to conduct on an industrial scale. Sodium abietate (Na-C20H29O2) was characterized by mass spectrometry, infrared spectroscopy, elemental analysis, X-ray diffraction, scanning electron microscopy, and energy-dispersive spectroscopy. To perform the antimicrobial tests, the determination of minimum inhibitory concentration and the disk diffusion assay was performed. The results obtained showed that the salt Na abietate performed an antimicrobial action against the bacterial strains *S. aureus*, *E. coli*, *L.monocytogenes*, and *S. enterica* Typhimurium and the yeast *C. albicans*. The disk diffusion test showed a high inhibition potential against *S. enterica* compared to the standard antimicrobial tetracycline, as an inhibition index of 1.17 was found. For the other bacterial strains, the inhibition values were above 40%. The MIC test showed promising results in the inhibition of *E. coli*, *L. monocytogenes*, and *C. albicans*, indicating bacteriostatic activity against the first microorganism and bactericidal and fungicidal activities against the others. Therefore, the results showed the action of Na abietate as a possible effective antimicrobial drug, highlighting its sustainability within a circular economy.

## 1. Introduction

In recent years, the World Health Organization (WHO) has been warning society about the increase in multidrug-resistant bacteria. According to the WHO, more than 50% of bacterial infections are becoming resistant to antibiotics, harming medicine, and putting lives at risk. The WHO says that in this way we are heading towards a post-antibiotic era, where common infections can kill again. Some examples are pneumonia, tuberculosis, and blood poisoning, which are becoming more difficult to treat as antibiotics become less effective [1]. A frequent question is to what extent the increase in resistance in these microorganisms is related to the treatments carried out during the COVID-19 pandemic, as there was an increase in the use of antimicrobial and general-purpose drugs [2,3]. The scarcity of new drugs effective against these microorganisms is another point to be highlighted, a problem generated by scientific and economic obstacles [4]. Due to this concern, it is very important to study and discover new drugs with antimicrobial action.

Abietic acid (C19H29COOH) is a diterpenoid that belongs to the terpene class and is found mainly in pine trees. Some works have already reported its antibacterial and antifungal efficacy; thus, abietic acid and its derivatives have an important potential as an alternative for new drugs, to be applied against antimicrobial resistance [5,6,7,8,9,10].

This work proposes the use of an organic sodium salt having as a precursor the abietate ligand (C19H29COO-) derived from abietic acid, one of the constituents of resins, such as that of *Pinus elliottii* var. *elliotti*. The preparation of Na abietate was obtained from an eco-friendly, low-cost synthesis route with good product yield and a reaction yield above 90% (m/m). The sodium abietate salt, Na(C19H29COO), was structurally characterized by X-ray diffraction, infrared spectroscopy, elemental analysis (CHN), scanning electron microscopy, energy-dispersive X-ray spectroscopy, visible spectroscopy (VISIBLE).

For the intended use as an antimicrobial agent, in vitro tests were performed using the Minimum Inhibitory Concentration (MIC), Minimum Bactericidal Concentration (MBC), Minimum Fungicidal Concentration (MFC), and disk diffusion methods. The satisfactory and reliable results we obtained support the design of future in vivo tests.

## 2. Results and Discussion

In this work, we present a detailed characterization of the sodium abietate salt Na-abietate. The salt was recrystallized from dissolution in warm water (40 °C) at the concentration of 200 g/L. The recrystallized salt was characterized and applied as an antimicrobial agent against fungi and bacteria. The salt has been used as a precursor ligand in the preparation of coordination compounds with transition metals [9,10]. The resulting compounds have different colors depending on the metal used, as well as antimicrobial properties when used as pigments in commercial white paint. We think that this work will give the reader a greater understanding of using a natural and sustainable binder, as it is extracted from the resin of reforestation trees.

### 2.1. Chemical Composition (CHN, ED, S and MS)

Experimental elemental analysis found 64.33% of carbon and 9.13% of hydrogen (Table 1). The experimental C/H ratio was 7.04, against a theoretical value of 8.22; the difference may be related to the presence of hydration water molecules, as sodium abietate is hygroscopic. Elemental analysis by energy-dispersive X-ray spectroscopy (EDS) was performed qualitatively to identify, in particular, the presence of sodium ions and adjust the salt composition. In the EDS spectrum (Figure 1), C, O, and Na atoms were found, as expected, since H atoms are not detectable.

The Na abietate mass spectrum (Figure 2a) was obtained in a dichloromethane (DCM) solution diluted in methanol. The most intense peak was located at m/z 1–301.21, equivalent to the theoretical molecular mass of deprotonated abietic acid. In the m/z range from 1–299 to 303 there were five peaks (Figure 2b) at the following m/z values (intensity): peak (1) 299.2011 (300,582), (2) 300.2042 (66,073), (3) 301.2172 (994,784); (4) 302.2205 (219,122); (5) 303.2171 (38,602), which represent the structure of protonated or deprotonated abietic acid, in an equivalent of up to two hydrogen atoms.

### 2.2. Vibrational Spectroscopy (FTIR)

In the FTIR spectrum (Figure 3), the bands attributed to the asymmetric (1544 cm^−1^) and symmetric (1397 cm^−1^) stretching of the carboxylate group (COO^−^) were prominent and are characteristic of the abietic acid present in the resin that resulted in the formation of the abietate group. The bands at 2926 and 2865 cm^−1^ were attributed to C–H binding [11,12], and those at 1450 and 1356 cm^−1^ to the symmetric elongation vibration modes of the CO_2_ molecule [11,12]. The band at 882 cm^−1^ was attributed to the C1–H deformation of the residual abietic acid [12].

For comparison purposes, the theoretical vibrational spectrum of Na abietate (Figure 4a) was obtained from DFT optimizations and frequency calculations performed in the ORCA software [13,14], using the AVOGADRO visualization software [15], which contains the Courier transform implemented in its algorithm. The quantum calculations were performed with B3LYP/6-311G(2d,2p) level of theory. The optimized geometry is shown in Figure 4b, with the labels used in the discussion.

The structure in Figure 4b shows that the 3D structure with three closed carbonic structures is stable and was improved by the computational calculations. In addition, the classical structure of an organic salt can be seen, where the cation sodium is bounded by carboxylic oxygen from the abietic acid molecule.

Important data obtained from computational calculations are reported in Figure 4b, which can indicate the approximated width and length of the molecule. In this Figure, the distance from the carbon atom labeled as 1 and carbon 6 is 3.9 Å and could be considered the width of the structure. On the other hand, the distance between the sodium atom and carbon 20 is 10.63 Å, indicating the possible length of the salt molecule. For the length of the anion, this corresponds to the distance between oxygen 1 and carbon 20 and is 10.35 Å.

The theoretical spectrum shows the presence of the bands attributed to the COO^−^ group (Table 2), highlighting a small difference (3 cm^−1^) compared to the experimental spectrum. Considering the experimental spectral resolution (4 cm^−1^) and the theoretical calculation, whose error was around 2 cm^−1^, we can consider the difference between the bands assigned to the carboxyl group to be acceptable and reliable. On the other hand, the difference between the bands attributed to [O–H] stretching was larger due to the presence of hydrogen bonds between abietate molecules in the experimental spectrum, whereas the single-molecule DFT calculations did not consider intermolecular van der Walls interactions or hydrogen bonds [16].

### 2.3. Ionic Properties and Zeta Potential (ζ)

Aqueous solutions of the sodium abietate salt were prepared with the purpose of measuring their ionic properties. The data are summarized in Table 3. A small increase in pH and conductivity can be seen with increasing concentration. The conductivity was attributed to the salt dissociation into the Na^+^ and (abietate^−^) ionic species. The increase in the concentration of sodium ions was responsible for the alkaline character of the medium, leading to a pH greater than 9.

After understanding the ionic properties of the Na abietate aqueous solutions, measurements of zeta potential and dynamic light scattering were carried out. For these measurements, an aqueous solution must be at a concentration of 1.0 × 10^−6^ mol L^−1^. Thus, starting from the 1.54 × 10^−3^ mol L^−1^ solution, dilutions were carried out that satisfied the parameters of the spectrophotometer that measured the light scattering properties. The measurements were performed in triplicate (Table 4), and it was possible to measure the average diameter of the abietate particles, that was found to be around 243.42 nm. The zeta potential averaged −44.5 mV, with an average mobility of −3.487 μm cm/Vs and a conductivity of 0.1846 mS/cm. The data summarized in Table 4 are consistent with conductive saline solutions and corroborate the data in Table 3.

### 2.4. X-ray Diffractometry (XRD)

The X-ray diffraction profile (XRDP) of the Na abietate salt is shown in Figure 5. No matching files were found in the Inorganic Crystal Structure Database (ICSD), International Centre for Diffraction Data (ICDD), and Crystallography Open Database (COD). The initial analysis of the X-ray diffraction profile showed that Na abietate has a structure characteristic of a non-crystalline molecular substance [9,10], with broad diffraction peaks and low intensity. The degree of crystallinity was found to be around 52%, with a crystallite size of 21.9 Å.

After recrystallization, the XRDP showed an intense peak at 2θ 15.51° with a d-value equal to 5.7 Å and a second peak at 2θ 28.84^o^ with a d value equal to 3.0 Å (Figure 5). However, the crystallite size remained of 22.0 Å, and crystallinity at 57.4%, considering the 2θ range between 7° and 25°. Extending the range to 7° and 50°, the crystallite size values were 19.7 Å, and the crystallinity decreased to 37.7%. Recrystallization favored a small increase in crystallinity without, however, altering the crystallite size of the Na abietate salt.

### 2.5. Antimicrobial Tests

#### 2.5.1. MIC/MBC/MFC Methods

Preliminary tests carried out in vitro to examine if that the Na abietate salt had promising activity in the inhibition of bacteria and fungi. In the minimum inhibitory concentration test (MIC) of a Na abietate solution, there was inhibition of the growth of the Gram-negative strain *Escherichia coli* starting from a Na abietate concentration of 7.8 μmol/L. For the Gram-positive strain *Listeria monocytogenes*, the minimum inhibitory concentration was 1000 μmol/L, the same that inhibited the growth of the yeast *Candida albicans*.

On the other hand, in the MBC and MFC tests, we did not observe the growth of colonies of *L. monocytogenes* and *C. albicans* in the presence of a Na abietate solution at a concentration of 1000 μmol/L. Therefore, the salt has bactericidal and fungicidal activity against these two strains, and bacteriostatic activity against *E. coli*.

Studies involving abietic acid (AA), such as that by Helfenstein et al. [6], also showed good antimicrobial properties. In that study, a concentration of 6 to 76 μg/mL inhibited the growth of *S. aureus* (ATCC 25923); for *E. coli* (ATCC 25922), the values ranged from 3 to 11 μg/mL, while a concentration between 20 and 39 μg/mL inhibited the growth of the yeast *C. albicans* (ATCC 90028). For *S. enterica* Typhimurium (ATCC 13311), the values varied from 31 to 125 μg/mL, considering 10 compounds derived from abietic acid. The work by Kurnaz et al. [7] also presented the inhibition values, determined with the MIC tests, of three tricyclic compounds derived from abietic acid; for *S. aureus* (ATCC-29213), the concentration values were 7.8 and 31.3 μg/mL, whereas, for *E. coli* (ATCC-25922), they were 15.6, 250, and 125 µg/mL. In the work by Silva et al. [17], the MIC value for *S. aureus* (ATCC 25923) was 1024 μg/mL, and that for *E. coli* (ATCC 25922) was 64 µg/mL, indicating that abietic acid has antimicrobial properties.

Considering all these data, our results showed that Na abietate can be used against Gram-positive and Gram-negative bacteria and has the same MIC level as abietic acid; however, it would be possible to reduce the costs of the entire process by obtaining the acid directly from the pine resin.

#### 2.5.2. Disk Diffusion Method

In the disk diffusion assay, the inhibition halos of the Na abietate salt (Figure 6), used against Gram-positive bacterial strains (*S. aureus* and *L. monocytogenes*) and Gram-negative ones (*E. coli* and *S. enterica* Typhimurium), were measured. For all strains tested, the antibiotic Tetracycline was used as a control.

The mean values (in millimeters) showed the largest inhibition zone (11.31 ± 0.64 mm) for the *E. coli* strain, followed by *L. monocytogenes* (10.08 ± 0.46 mm), *S. aureus* (8.42 ± 0.84 mm), and *S. enterica* Typhimurium (7.25 ± 0.35 mm). 

The activity index [18,19,20,21,22,23] of Na abietate for each evaluated bacterial strain was estimated by the ratio between the inhibition zones of the tested compound and of the standard antimicrobial. Promising results were obtained, especially against the *S. enterica* Typhimurium strain, with an activity index of 1.17 (Figure 6d). A high inhibition index was also found for the bacteria *S. aureus* (*A.I* = 0.96), *E. coli* (*A.I* = 0.65), and *L. monocytogenes* (*A.I* = 0.45). However, the largest halo does not indicate the best inhibition, as observed in the activity calculation, where the *S. enterica* Typhimurium strain showed a higher percentage of inhibition than that measured with the antibiotic Tetracycline, used as a control for Na abietate.

Studies such as that of Cabezas-Pizarro et al. [24] showed that sodium compounds with aliphatic- and aromatic-chain carboxylic acids have a good antimicrobial activity, compared with potassium salts. Furthermore, their bioactivity can be related to the number of carbon atoms in the molecule, with compounds with shorter carbon chains having a higher antimicrobial activity, but it tends to increase with the increasing chain length until a shear phenomenon occurs. For *L. monocytogenes* (ATCC 19116) and *C. albicans* (ATCC 10231), for example, the inhibition values of the compound sodium octanoate were 25 mg/mL and 100 mg/mL, respectively, while for potassium octanoate, the inhibitory concentrations were of 50 and 100 mg/mL for the same strains. This shows that the sodium metal center influences the antimicrobial potential, as seen in this work.

## 3. Material and Methods

### 3.1. Reagents

The following reagents were purchased and used without any additional treatment: sodium hydroxide (NaOH, Neon^®^, P.A.), ethanol (fuel), Mueller Hinton agar, broth (Kasvi), Sabouraud dextrose broth (TM MEDIA), potato dextrose agar (Himedia), TTC dye (Sigma-Aldric). The pine resin was collected in nature and was purified according to the process described in this work (Item 3.2).

### 3.2. Na abietate Synthesis

The Pinus resin was extracted from reforested trees, which were seven years old. For this, cuts were made in the stem of the trees so that the resin drained from its interior. The resin is part of the defense mechanism of the trees. In most cases, a sulfuric acid-based paste was applied to the cuts to promote greater resin production.

The resin was collected in plastic bags fixed next to the cuts made in the tree. These bags were exposed to weather, rain, sun, insects, leaves, and other debris. Because of this, the first step consisted in washing the resin with water to remove particulates and sulfuric acid, controlling it until it reached pH 7. Then, the resin was solubilized in fuel ethanol (95%) and filtered to remove residual particles (Figure 7). In a rotary evaporator, the solvent was recovered, and the purified resin was obtained.

The abietate sodium salt synthesis process was carried out by homogenizing the Pinus resin and sodium hydroxide solution (0.75 mol L^−1^) in a beaker. The ratio used was 1:1 (m:m). Then, the solution was kept under agitation and heated to 60 °C until total evaporation of the water.

The residual solid in the beaker was kept in an oven at 70 °C for 3 h. Then, the solid was macerated and dissolved to carry out salt purification. Thus, 20 g of Na abietate was added to 100 g of water and kept under stirring for 12 h. It was then left to rest in a refrigerator (4 °C), and precipitation of the recrystallized salt occurred. After this, it was filtered through a funnel with a porous plate (number 2) and kept in a desiccator for 48 h. Thus, the solid was macerated, sieved, and used in the characterizations and applied tests. Figure 7 shows the resin purification steps, the transformation, and the recrystallization of the Na abietate salt.

### 3.3. Characterization

#### 3.3.1. Composition by CHN, MS, and EDS

Three techniques were used to estimate the elemental composition of Na abietate: elemental analysis of CHN elements, mass spectroscopy, and energy-dispersive X-ray spectroscopy. The equipment used to perform the elemental analysis (CHN) was a PerkinElmer 2400 Series II system; energy dispersive X-ray spectroscopy (EDS) was performed coupled to scanning electron microscopy (SEM) using a model TM3000, Hitachi, system. These two techniques were performed to estimate the composition of the samples.

The mass spectrum was used to identify mass/charge ratios and determine the structure of Na abietate. For this, a solution of Na abietate in dichloromethane was prepared and diluted in methanol, then injected in a Bruker Amazon Speed ETD system, ion trap (MS-MS) with low resolution, in negative electrospray ionization. Conditions: the drying gas flow (4 L min^−1^) was used at a temperature of 200 °C, nitrogen as a nebulizer gas under a pressure of 7 psi, and the voltage was 4500 V.

#### 3.3.2. Vibrational Spectroscopy (FTIR)

The FTIR spectrum was obtained with a PerkinElmer Frontier spectrophotometer, in the region of 4000–650 cm^−1^, in the attenuated total reflectance (ATR) acquisition mode, with a high-capacity ZnSe crystal for analyzing powders, solids, and liquids.

#### 3.3.3. Zeta Potential (ζ)

The potential determination was carried out for an aqueous solution of Na abietate, without the addition of KCl electrolyte, using a Nano-ZS90 Zetasizer Nano series system from Malvern.

#### 3.3.4. X-ray Diffractometry (XRD)

The X-ray diffractogram was obtained on a Bruker X-ray diffractometer, model D2 Phaser, operating under the conditions of copper cathode (λ = 1.5418 Å), 30 kV potential, 10 mA current, 2θ degree between 7° and 60°, and increment of 0.07°/seg.

### 3.4. Antimicrobiological Tests

#### 3.4.1. Minimum Inhibitory Concentration (MIC)

The biological Minimal Inhibitory Concentration test was performed for Gram-positive *Listeria monocytogenes* (ATCC 19111) and Gram-negative *Escherichia coli* (ATCC 25922) bacteria. Additionally, the compound was also tested against the *Candida albicans* (ATCC 10231) fungus. The methodology used was that of the Clinical and Laboratory Standards Institute (CLSI) [25,26], adapted from the work of [10].

The microbial inoculum was adjusted to a concentration of 1.5 × 10^8^ CFU mL^−1^ using a 0.5 McFarland scale [20]. The test was performed in a 96-well plate in quadruplicate, following the serial microdilution method. The applied concentrations were 1000, 500, 250, 125, 62.5, 31.2, 15.6, and 7.8 μmol L^−1^. The culture medium used for the bacteria was Mueller–Hinton (MH) broth, and that for the fungus was Sabourand dextrose broth. A Na-abietate solution (2 mmol L^−1^) was added to well “A” and diluted up to well “H” as described in [10]. Then, the microorganism inoculum was added. As a negative control, 100 μL of MH broth and ethanol was used, and the positive control was the antibiotic Tetracycline (1000 μg mL^−1^) for the bacteria and a Fluconazole 10% solution for the yeast. The plate was incubated for 24 h at 37 °C. Finally, the TTC dye (2,3,5-triphenyl tetrazolium chloride; 0.125%) was added to the reaction mixture, and incubation was continued for another two hours. The addition of the dye indicated the presence of living or non-living microorganisms in the medium through coloration. A reddish-pink color indicated the presence of active bacteria and fungus in the wells; if there was no coloration, the microorganisms had been inactivated.

#### 3.4.2. Minimum Bactericidal Concentration (MBC) and Minimum Fungicidal Concentration (MFC)

The MBC and MFC tests were performed next, based on a previous work [10]. The mixture that inhibited the growth of the microorganisms in the MIC test was transferred to a Petri dish. For the bacteria, the medium used was MH agar, and incubation was carried out at 37 °C for 24 h; for the fungus, the Sabourand dextrose agar was used, with incubation at 28 °C for 48 h. The interpretation of the MBC was performed by analyzing the growth (bacteriostatic effect) or lack of growth (bactericidal effect) of the samples tested on the plates, and for MFC, the evaluation consisted in observing the growth of the microorganism on the plate (fungistatic effect) or the lack of growth (fungicidal effect) [10].

#### 3.4.3. Disk Diffusion Method

In the disk diffusion test, the Gram-positive strains *Staphylococcus aureus* (ATCC 25923) and *Listeria monocytogenes* (ATCC 19111) and the Gram-negative strains *Escherichia coli* (ATCC 25922) and *Salmonella enterica* Typhimurium (ATCC 0028) were tested. The methodology was adapted according to the literature [10,19].

For the test, sterile filter paper disks 5 mm in diameter were prepared and placed in a Petri dish, and 50 μL of Na abietate solution (2 mmol L^−1^ in 10% ethanol) was added. A control disk containing Tetracycline (Sensibiodisc CECON-30 mcg) antibiotic was added to each plate. The assays were performed in duplicate per plate, with solvent (10% ethanol) and Tetracycline controls. Two plates were tested per sample, that is, the experiment was conducted in quadruplicate. The results were obtained by measuring the halo of inhibition caused by each disk, and then an average of the data was calculated.

The activity index (*A.I*) was also calculated according to Equation (1) [18,19,20,21,22,23].
(1)$$A.I=\frac{Inhibition\;zone\;\left(mm\right)}{Inhibition\;zone\;of\;standart\;drug\;\left(mm\right)}$$

## 4. Conclusions

The simple purification of pine resin allowed its transformation into the sodium abietate salt. After its recrystallization, it was possible to determine the chemical composition of the salt by mass spectrometry and a chemical analysis, resulting in establishing the molecular formula Na(C19H29COO), corresponding to Na abietate. The vibrational spectrum showed the characteristic bands of the carboxylate group (COO^−^) and the C-H bonding. A DFT study made it possible to generate a theoretical vibrational spectrum corroborating the experimental data. Measurements of the zeta potential and ionic properties indicated a negative charge inherent in the dissociation of the Na abietate salt in water, with an alkaline pH (approximately 9.5). The crystallite size (22 Å) and crystallinity (54%) were calculated by XRD. The sodium abietate salt allows to solubilize the “resin” in water and can be applied in antimicrobial solutions. In vitro tests showed good responses against Gram-positive and Gram-negative strains of bacteria, compared to the Tetracycline antibiotic used as a control. The water solubility and antifungal activity of sodium abietate are important for the use of this salt against phytopathogenic fungi.

## Figures and Tables

**Figure 1 antibiotics-12-00514-f001:**
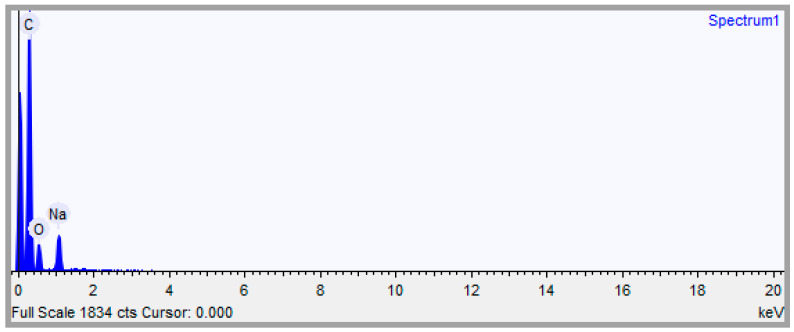
EDS spectrum. Conditions: acquisition time, 20.0 s; accelerating voltage, 15.0 kV.

**Figure 2 antibiotics-12-00514-f002:**
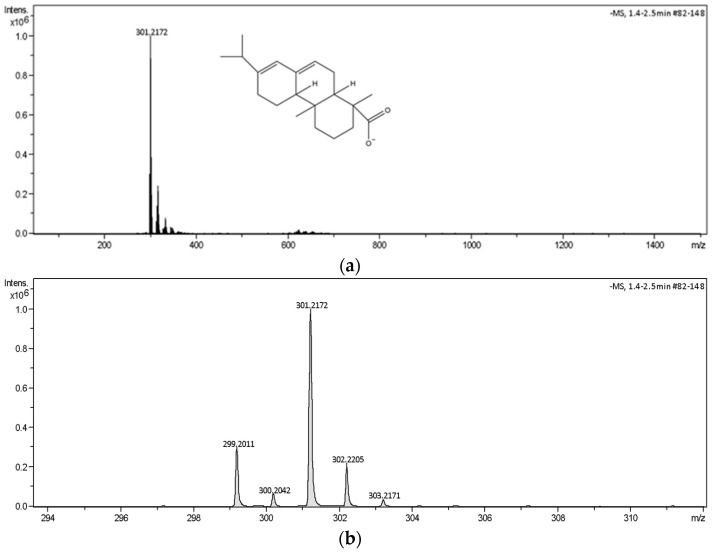
Mass spectra of Na abietate the with the molecular structure of the abietate ion (**a**). More intense peaks in the m/z range from 1–298 to 1–306 correspond to the protonated and deprotonated forms of abietic acid (**b**).

**Figure 3 antibiotics-12-00514-f003:**
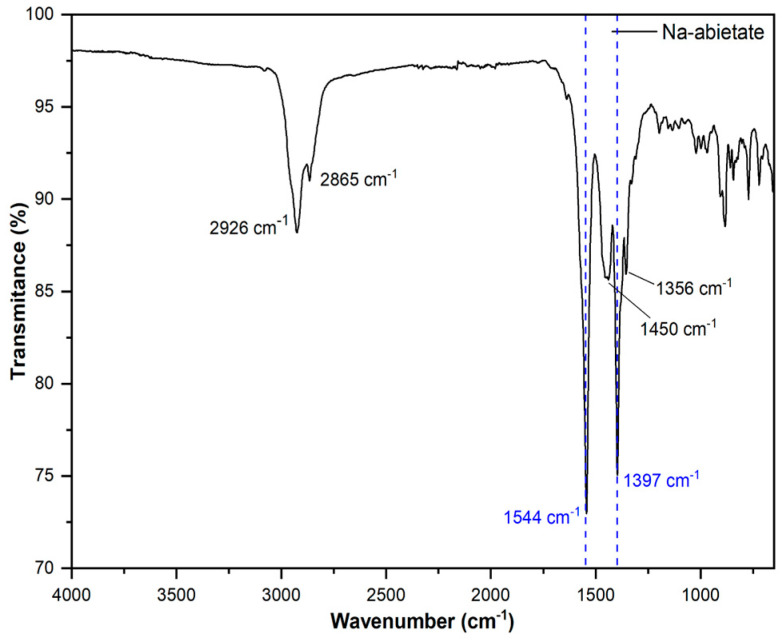
FTIR spectrum of the powdered Na abietate sample, obtained in ATR mode, without the need to prepare a KBr pellet.

**Figure 4 antibiotics-12-00514-f004:**
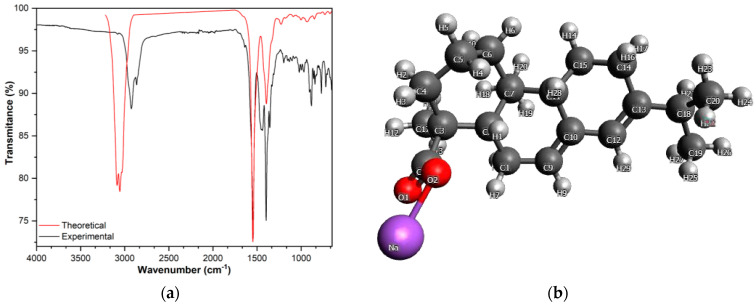
Theoretical and experimental infrared spectrum (**a**) and optimized structure on Na abietate in B3LYP/6-311G(2d,2p) level (**b**).

**Figure 5 antibiotics-12-00514-f005:**
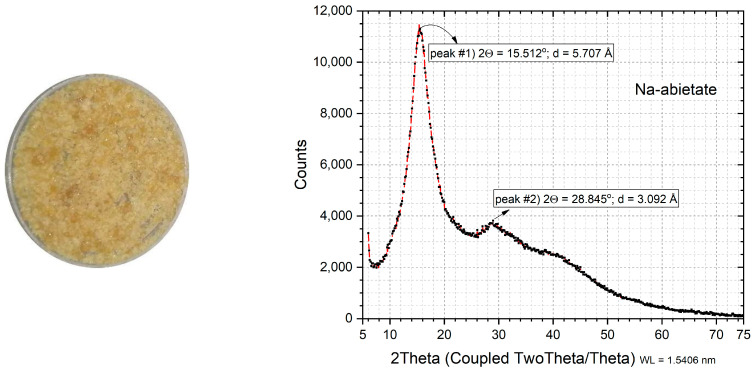
XRD profile of Na abietate. Image of the sample holder containing the powdered sample.

**Figure 6 antibiotics-12-00514-f006:**
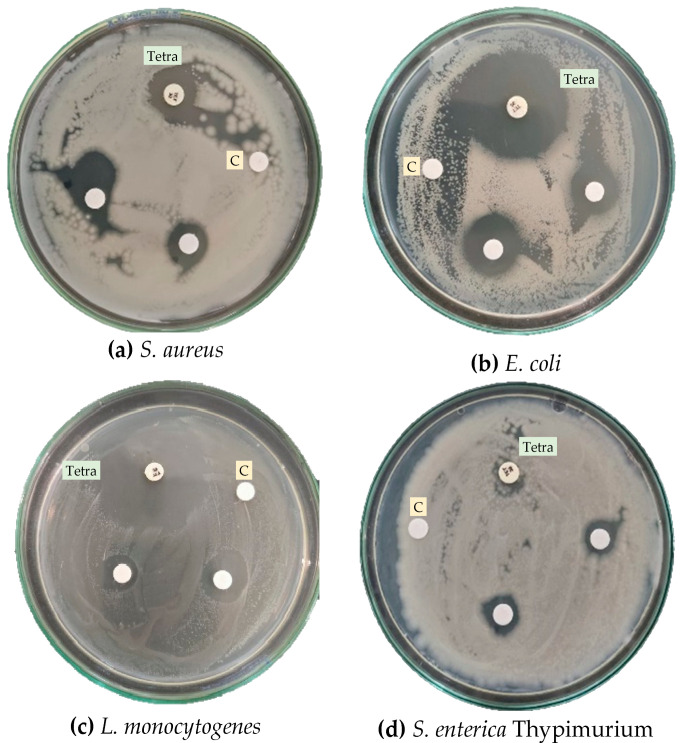
Disk diffusion test results against (**a**) *Staphylococcus aureus*, (**b**) *Escherichia coli*, (**c**) *Listeria monocytogenes*, and (**d**) *Salmonella enterica* Typhimurium. In the images, “Tetra” represents the Tetracycline antibiotic, and “C” represents 10% ethanol, as controls.

**Figure 7 antibiotics-12-00514-f007:**
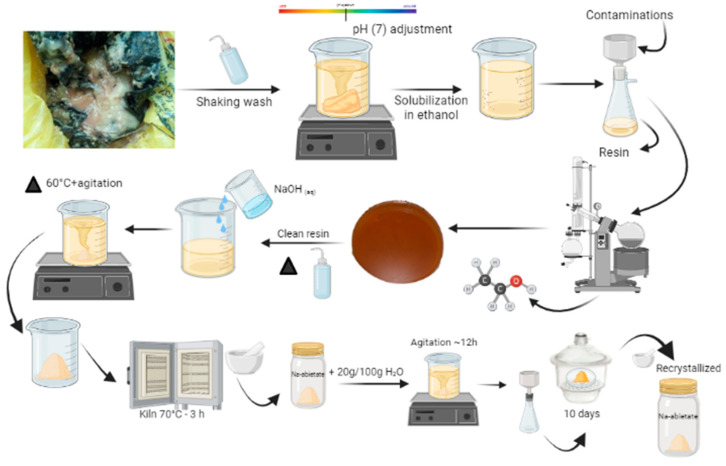
Resin cleaning and Na abietate salt synthesis. Created in BioRender.com.

**Table 1 antibiotics-12-00514-t001:** Elemental composition of Na(C_20_H_29_O_2_).

	Sodium (Na)	Carbon (C)	Hydrogen (H)	Oxygen (O)	C/H Ratio
Mass (%) theoretical	7.0861	74.0412	9.0096	9.8630	8.218
Mass (%) found	---	64.33	9.13	---	7.04
Composition by EDS	4.806	73.833	---	21.361	---

**Table 2 antibiotics-12-00514-t002:** Main vibrational bands and their attribution in theoretical and experimental FTIR spectra.

Theoretical (cm^−1^)	Experimental (cm^−1^)	Δ_Th-Ex_ (cm^−1^)	Assignments
3087	2926	161	ν O-H/νC-H (sp3) [11] *
3055	2865	190	ν O-H/ν C-H
1547	1544	3	ν_ass_ COO^−^ [10,16]
1394	1397	3	ν_sim_ COO^−^ [10,16]

* Adapted with permission from Ref. [11]. © 2018 Elsevier Ltd. All rights reserved.

**Table 3 antibiotics-12-00514-t003:** Ionic properties of Na abietate aqueous solutions, pH and conductivity measurements.

Mass (g)	Concentration (g mol^−1^)	pH	Conductivity (mS/cm)
0.5	1.54 × 10^−3^	9.76	132.7
1.0	3.08 × 10^−3^	9.87	140.6
2.0	6.16 × 10^−3^	9.93	150.6

**Table 4 antibiotics-12-00514-t004:** Measurements of zeta potential, particle size, and conductivity by DLS.

Sample	Size (d. nm)	ζ Potential (mV)	Mobility (μm cm/Vs)	Conductivity (mS/cm)
1	266.50	−43.5	−3.410	0.183
2	257.70	−45.9	−3.602	0.184
3	206.08	−44.0	−3.451	0.187
Average	243.42	−44.5	−3.487	0.1846

## Data Availability

Not applicable.

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
