# Peer review of "Green Synthesis of Na abietate Obtained from the Salification of Pinus elliottii Resin with Promising Antimicrobial Action"

_antibiotics, 2023, doi:10.3390/antibiotics12030514_

Round 1

Reviewer 1 Report

Review the attached file to see the comments, please.

Author Response

On behalf of the authors, I thank you for your criticisms and suggestions. Corrections were made directly to the manuscript and are highlighted in red. Attached are the specific responses.

Reviewer 2 Report

Dear authors

In the Abstract it was stated that 146% of S. enterica were inhibited. How it was concluded as the end of inhibition is 100%.

In figure 6, Gram-negative bacteria Listeria monocytogenes is wrong. Gram-positive should be instead. Also Candida albicans is italic. Please also check in the whole manuscript.

In figure 7, disks concentrations are unknown except for the control.

The cytotoxicity of the compound against normal eukaryotic/human cells has not been evaluated.

English writing needs improvement.

Best regards

Author Response

(The authors gave the same response as above.)

Reviewer 3 Report

Abietic acid, a well-known compound found in many pine resins and available from several dozens of commercial suppliers, has been comprehensively studied from both chemical and biological points of view for several decades. Several recent papers report studies of its mechanisms of particular biological activities. The manuscript of Schons et al. reports yet another isolation of abietic acid (in the form of Na salt) from a pine resin, its chemical characterization, and routine preliminary activity tests on several pathogens. The manuscript is completely lacking of scientific novelty. The activity tests are obsolete and suitable only for preliminary screening (the activity index A is not a reliable indicator of the antibiotic activity of a substance). Moreover, the description of activity tests is rather careless: no substance or tetracycline load per disc is given, some abbreviations (e.g. strains on Fig. 8) are missing, several pathogen strains are not characterized. In addition, the level of scientific presentation is low: figures 6 and 7 do not contain any useful information, there are two figures 8, a mix of styles in the references section, etc. Thus, this referee definitely recommends rejection.

Author Response

On behalf of the authors, we thank you for the criticism. Our comments and your criticisms are attached.

Round 2

Reviewer 3 Report

The revised version of the manuscript contains addition to 2.3.2 Disk diffusion method section, Materials and method section, as well as several new references. The additions do not change the major point – the lack of scientific novelty. The reported antimicrobial assays add nothing to known biological properties of abietate. Neither new targets/mechanisms, nor any useful applications were reported. It seems the data reported would be not of interest for Antibiotics readers.

Author Response

Author's Reply to the Review Report (Reviewer 3)

The revised version of the manuscript contains addition to 2.3.2 Disk diffusion method section, Materials and method section, as well as several new references. The additions do not change the major point – the lack of scientific novelty. The reported antimicrobial assays add nothing to known biological properties of abietate. Neither new targets/mechanisms, nor any useful applications were reported. It seems the data reported would be not of interest for Antibiotics readers.

Response: Dear Reviewer 3, thank you for the suggestion.

We performed a new search on the main search engines for scientific journals; and found a restricted number of articles using abietic acid as an antimicrobial agent, and in the form of Na-abietate salt, we did not find any that present characterization and application as shown in our manuscript.

The manuscripts use abietic acid (AA) purchased from a commercial supplier, with a high degree of purity, that is, with a high cost per gram of AA. Our proposal for the transformation of pine resin into Na-abietic salt goes beyond this manuscript, we use it as a coordenation ligand for the formation of complex compounds with antimicrobial properties associated with color, that is, a special pigment.

The complexes Zn-abietate (Dyes & Pigments, ref. 9) and V-abietate (Molecules, ref. 10) were prepared. The manuscripts describe the characterization and application as a pigment in building paint.

We appreciate your criticisms and suggestions, which helped to improve the final version of the manuscript.